# Exploration on Optimized Control Way of D-Amino Acid for Efficiently Mitigating Membrane Biofouling of Membrane Bioreactor

**DOI:** 10.3390/membranes11080612

**Published:** 2021-08-11

**Authors:** Zhan Gao, Zhihao Yu, Xiaoli Zhang, Shougang Fan, Huiyu Gao, Caini Liu, Qixing Zhou, Huaiqi Shao, Lan Wang, Xiaoyan Guo

**Affiliations:** 1Ministry of Education Key Laboratory of Pollution Processes and Environmental Criteria, Tianjin Key Laboratory of Environmental Technology for Complex Trans-Media Pollution, Tianjin Key Laboratory of Environmental Remediation and Pollution Control, College of Environmental Science and Engineering, Nankai University, No. 38 Tongyan Road, Tianjin 300350, China; 2120190533@mail.nankai.edu.cn (Z.G.); 1120190234@mail.nankai.edu.cn (Z.Y.); zhangxlfamily@126.com (X.Z.); 1120180195@mail.nankai.edu.cn (S.F.); 2120180636@mail.nankai.edu.cn (H.G.); 1120200243@mail.nankai.edu.cn (C.L.); zhouqx523@126.com (Q.Z.); envwangl@nankai.edu.cn (L.W.); 2College of Chemical Engineering and Material Science, Tianjin University of Science & Technology, No. 29 Thirteenth Street, TEDA, Tianjin 300457, China; shaohq@tust.edu.cn

**Keywords:** membrane bioreactor, membrane biofouling, D-amino acid, control way, model evaluation

## Abstract

The thorny issue of membrane biofouling in membrane bioreactors (MBR) calls for new effective control measures. Herein, D-amino acid (DAA) was employed to mediate MBR membrane biofouling by inhibiting biofilm information and disintegrating formed biofilm. Different DAA control ways involving membrane property, DAA-adding timing, and DAA-control mode were explored through experiments and the multiple linear regression model and the response surface methodology. The optimized DAA control ways were acquired, involving DAA used as an active agent, and the DAA-adding timing of 4 h cultured before running, as well as both hydrophilic and hydrophobic membrane, resulting in an approximately 40.24% decrease in the membrane biofouling rate in comparison with the conventional MBR. DAA is an efficient membrane biofouling mediating approach for MBR under optimized control ways combination and a facile solution for solving membrane biofouling in actual membrane systems.

## 1. Introduction

Membrane biofouling, resulting from biofilm forming on the membrane surface, is an inevitable and annoying phenomenon in the membrane bioreactor (MBR) [1,2,3]. It not only decreases membrane permeate productivity and quality but also increases operating costs owing to frequent membrane chemical cleaning, as well as shortens membrane lifespan [4,5,6,7]. Therefore, it is indispensable to develop a beneficial approach to alleviate membrane biofouling for more efficient application of MBR [8].

Biological control approaches have attracted considerable attention due to excellent biofouling mitigation efficiency without killing the deposited cells [9,10], thereby displaying great potential for biofouling control in MBR. Particularly, D-Tyrosine (as a typical D-amino acid (DAA)), a kind of newly discovered environmentally-friendly anti-biofouling biological molecule [11,12,13], can block cell wall synthesis through changing peptidoglycan composition and structure [14], and further trigger bacterial amyloid fibers to disassemble via incorporation into the cell wall [12], demonstrating promising anti-biofouling efficacies. For instance, D-tyrosine significantly inhibited the microbial attachment to hydrophilic glass and hydrophobic propylene surfaces [15]. In addition, DAA was reported to notably reduce biomass attachment on the nanofiltration 270 membrane at 30 min and 1 h, whereas it gained a diminishing anti-biofouling effect at more than 4 h of incubation [16]. Additionally, D-tyrosine not only inhibited microbial formation on membrane surfaces as an effective agent but also promoted biofilm detachment from biologically fouled membrane surfaces as a cleaning agent [10]. From the forgoing, when D-tyrosine was used to control membrane biofouling, these factors involving the membrane property, DAA-adding timing, and DAA-control mode, etc., may have important influences on membrane antifouling effects. However, this research has not been systematically studied.

Therefore, different membrane properties, DAA-adding timing, and DAA-control modes were selected to evaluate the membrane anti-biofouling performances of DAA, further the influence weights of these factors were analyzed, and then these factors were optimized, with the expectation of finding an efficient anti-biofouling approach for MBR.

## 2. Materials and Methods

### 2.1. Experimental Set-Up of MBR

In this experiment, the pilot-scale MBR (Tianjin Bohua Environmental Engineering Technology Co., Ltd., Tianjin, China) was set up to investigate the membrane anti-biofouling performances via DAA. The schematic diagrams and real photos are illustrated in Figure 1 and Appendix A, respectively. The MBR had a chamber with volumes of 43 L, and hollow fiber membranes with hydrophobic and hydrophilic properties, respectively, represented in Table 1. The MBR was seeded with activated sludge collected from a local sewage treatment plant (Beitang sewage treatment plant in Tianjin, China) and fed with synthetic municipal wastewater (the key compositions are shown in Appendix A) [17]. The mixed liquor suspended solids (MLSS) concentration in the reactor was maintained at 4900–5100 mg·L^−1^ by the discharge of excess sludge. Ambient temperature was operated at under 25 ± 2 °C, and pH was controlled at the range of 7.0–7.5. Then, the MBR was operated in the following mode: all the membrane modules ran for 8 min and stopped for 2 min; and the membrane permeate flux was maintained at a constant of 1.2 L·m^−2^ h^−1^, which corresponded to a hydraulic retention time (HRT) of 10 h, and an aeration rate of 40 L· min^−1^ was continuously supplied for microbial growth and mediating membrane fouling, as well. The trans-membrane pressure (TMP) of each membrane module was continuously monitored by manometers and recorded by a recorder every 1 h for evaluating the evolution of membrane biofouling [18]. Once the TMP reached 30 kPa, the membrane module was taken out for physical cleaning. Firstly, the microorganisms adhered to the membrane were gently scraped off to prepare the mixed solution for extracting the extracellular polymeric substances (EPS) to determine the content of protein and polysaccharide. Secondly, these membrane modules were soaked in pure water for 1 h to remove the fouling. Afterwards, the physically cleaned membrane modules were installed for the next cycle. In this way, three cycles were operated for the MBR. All the above parameters were automatically controlled by the programmable logic controller (PLC) system (designed by Tianjin Bohua Environmental Technology Engineering Co., Ltd., Tianjin, China). In addition, D-tyrosine (supplied by Aladdin Reagent Co., Ltd., Shanghai, China) was added into the chamber of the MBR with a concentration of 6 mg·L^−1^, which is the minimum and appropriate concentration that D-tyrosine strongly inhibits biomass attachment and biofilm formation on the membrane surface [19]. Different DAA control ways will be dealt with in Section 2.2. Detailed information on operational parameters can be found in Appendix A.

### 2.2. Control Protocols of DAA

DAA, as a kind of novel anti-biofouling agent, was used to mediate MBR membrane biofouling in different ways, as shown in Appendix A. DAA was, respectively, used as an active agent and cleaning agent to investigate the anti-biofouling performance of hydrophilic and hydrophobic membranes. In the meantime, the adding timing of DAA was also evaluated before and after running of MBR, respectively; the specific operating conditions are explained in Table 2.

### 2.3. Analytical Methods

Changes in TMP of MBR with different DAA control ways were monitored by PLC systems throughout the experiment to assess membrane biofouling behaviors [20]. Generally, the membrane module was considered fouled when the normalized TMP (against baseline) reached 30 kPa under the constant flux operation mode [5]. Further, the membrane biofouling extents can be analyzed by EPS concentration of sludge on membrane surface for the fouled membranes [21], which may be obtained by measuring the sum contents of polysaccharides and protein via the phenol sulfuric acid method and Coomassie brilliant blue method, respectively [1,22]. Additionally, the effluent quality of MBR with different DAA control ways was determined by total organic carbon (TOC) analyzer (multi N/C3100, Jena Analytical Instrument Co., Ltd., Jena, Germany).

### 2.4. Optimization Approaches

In order to quantitatively unravel the relationship of the anti-biofouling efficacies of MBR and various DAA control ways, the multiple linear regression (MLR) model [1,23] was introduced to quantify the individual and interactive contributions of membrane property, DAA-adding timing, and DAA-control mode to the membrane biofouling. Among them, the membrane biofouling rate $r_{p}$ (kPa·h^−1^) was regarded as the dependent variable, which can be calculated according to the following equation:(1)$$\;r_{p}=\frac{TMP_{max}-TMP_{min}}{\Delta t},$$
where $TMP_{max}$ and $TMP_{min}$ (kPa) are the maximum and minimum TMP in one filtration-physical cleaning cycle, and Δt (h) is the duration of the cycle.

The DAA control ways (represented as membrane property, DAA-adding timing, and DAA-control mode) were taken as the independent variables. Thus, the relationship between biofouling rate $r_{p}$ and DAA control ways was described by MLR as the following formula:(2)$$E\left(y\right)=f\left(x_{1},\ldots ,\;x_{m}\right)\approx \;b_{0\;}+\sum_{i=1}^{m}b_{i}\;x_{i}+\sum_{i=1}^{m}\sum_{j=1}^{m}b_{ij}x_{i}x_{j}+\varepsilon \;,$$
where *E*(*y*) represented the biofouling rate $r_{p}$; $x_{1}$, $x_{2}$, …, $x_{m}$ stood for membrane property, DAA-adding timing, and DAA-control mode, respectively; $b_{0}$ was the constant factor; $b_{i}$, $b_{ij}$ referred to the coefficient of liner and interaction effects, respectively; *ε* referred to the random error.

Afterwards, analyses of variance (ANOVA) and hypothesis testing for the regression coefficient (*t*-test) were also performed to determine the apparent correlations between the two variables at a 95% confidence interval (*p* < 0.05 in Pearson correlation), thus identifying the major factors affecting the membrane biofouling. All statistical analyses were carried out using SPSS software 22.0 [24].

The weight analysis of DAA control ways was clear by using the MLR model, which points out the direction for further optimizing and predicting precisely. Herein, the various DAA control ways combination can be optimized and predicted by the response surface methodology (RSM) [25,26] to obtain better anti-biofouling efficacies. In this study, a Box-Behnken design (BBD)-based RSM optimization was conducted with Design-Expert 8.0 software, in which a three-factor two-level design was employed, as shown in Table 3. The membrane property ($x_{1}$), DAA-adding timing ($x_{2}$), and DAA-control mode ($x_{3}$) represented three factors and were given two code levels, respectively, and the dependent variable R was still the membrane biofouling rate $r_{p}$. Afterwards, the least-squares regression method was used to analyze the results to predict the process response [26]. Besides, the comparision between the simulated and the actual value was conducted to confirm the accuracy of the RSM model. The above quantitative analysis and optimization and prediction were all based on the results of preliminary experiments.

## 3. Results

### 3.1. Anti-Biofouling Performances of MBR with DAA Control Way

The anti-biofouling performance of MBR with DAA control way is represented in Figure 2, in the case of using DAA as a cleaning agent to control membrane biofouling of hydrophilic PAN membrane, with that of conventional MBR as control. Delightedly, it can be obviously seen that the running time of the MBR with DAA control way was longer than that of conventional MBR in Figure 2a, which indicated that the anti-biofouling capability of MBR with DAA control way was superior to that of conventional MBR. Further, the anti-biofouling abilities were calculated in every biofouling-physical cleaning cycle according to Equation (1), as represented in Appendix A. Obviously, the average membrane biofouling rate of 1.07 kPa·h^−1^ in the MBR with DAA control way was lower 21.90% than that of 1.37 kPa·h^−1^ in conventional MBR. These results demonstrated that the addition of DAA could be a useful approach to mediate membrane biofouling in MBR, based on the inhibition of EPS production shown in Figure 2b, which was consistent with lab-scale studies of Malaeb et al. [10,27]. As shown in Figure 2b, the EPS content of sludge on membrane surface in MBR with DAA control way was reduced by approximately 22.31% compared to conventional MBR, which was resulted from the inhibition and disassembly impact of DAA for biofilm on membrane [10]. It should be noted that the protein content in MBR with DAA control way was even higher than that in conventional MBR, although the total EPS content was lower. Verifiably, the increased protein content was revealed to bits of DAA (as a kind of protein) adhered to the membrane surface, according to the reduction of average TOC removal rate from MBR effluent, compared with that of conventional MBR (Appendix A). This is to say, DAA can effectively trigger biofilm disassembly or inhibit biofilm formation to retard membrane biofouling in MBR with DAA control way, through incorporation into cell wall to disassemble the bridged bacterial amyloid fibers [12], even though some of DAA may adhere to the membrane surface. Nonetheless, the effluent qualities of MBR with DAA control way and conventional MBR were very close, as shown in Appendix A, and DAA had almost little effect on the effluent quality of MBR with DAA control way. In addition, the anti-biofouling performances of MBR with DAA control way represented in Figure 2 involved a kind of membrane biofouling control way of DAA, which may further enhance the membrane anti-biofouling tendencies by regulating the interaction way of DAA with biofilms. Hence, the influence of other DAA control ways on the anti-biofouling efficacies of MBR will be further investigated as follows.

### 3.2. Influences of DAA Control Ways on Anti-Biofouling Performances of MBR

The various DAA control ways, including membrane property, DAA-adding timing, and DAA-control mode, as shown in Appendix A, may have important influences on anti-biofouling performances of MBR, which will be, respectively, investigated to explore the strategies on improving anti-biofouling performances of MBR.

#### 3.2.1. Effects of Membrane Property

The different membrane properties involving hydrophilicity and hydrophobicity have significant impacts on membrane biofouling control [28,29]. The three-cycle membrane biofouling behaviors of MBR with hydrophilic and hydrophobic membrane modules are represented in Figure 3a. It can be seen that the running time of three cycles for hydrophilic MBR was markedly longer than that of hydrophobic ones. Meanwhile, after every physical cleaning, the recovered initial TMP of about 5 kPa for hydrophilic MBR was obviously lower than that of 10–12 kPa for hydrophobic MBR. These results meant that the hydrophobic MBR was more prone to rapid biofouling and involved physically irreversible biofouling. In other words, the hydrophilic PAN membrane combined with DAA was more efficient to inhibit membrane biofouling than hydrophobic PVDF membrane with DAA. The higher anti-biofouling performance of the PAN membrane was due to the formation of a hydrated layer on the membrane surface, which can prevent the deposition and adsorption of EPS [2]. As shown in Figure 3b, the total EPS amount was 40.33% lower in the hydrophilic PAN MBR, compared with the hydrophobic PVDF MBR. It should be noted that DAA is hydrophilic so that it is not easy to adhere to the hydrophilic PAN membrane surface, thus contributing fewer EPS contents in hydrophilic PAN membrane surface. Therefore, the higher anti-biofouling performances of hydrophilic MBR can be attributed to the combination effects of the efficient inhibition biofilm formation of DAA and the effective hydrophilicity of PAN.

#### 3.2.2. Effects of DAA-Adding Timing

Considering that a certain amount of contact timing is required for the interaction of DAA with microorganisms, different DAA-adding timings will be explored to ascertain the effects of DAA-adding timing on the anti-biofouling performances of MBR in case of hydrophilic PAN membrane. As shown in Figure 3c, the running time to reach 30 kPa of TMP for the MBR with DAA control way before feeding was remarkably increased and nearly 1.35 times longer than that for the MBR by adding DAA after feeding. It suggested that the onset of biofouling has been delayed by around 26% of running time by adding DAA before running. Meanwhile, the total EPS amount for the MBR with adding DAA before feeding was also 11.36% lower than that for the MBR by adding DAA after feeding, as represented in Figure 3d. It can be inferred that DAA can more effectively inhibit the formation of biofilm by reducing the production of EPS with the increase of DAA interaction time with microorganism, meanwhile avoiding the adhesion of DAA on the membrane surface, thus efficiently mediating the membrane biofouling [30]. Hence, it is beneficial for controlling the membrane biofouling by pre-adding DAA to effectively make contact with microorganisms.

#### 3.2.3. Effects of DAA-Control Mode

In view of the two different interaction modes of DAA with microorganism, the inhibition biofilm formation and the disintegration resulting biofilm [10,15], two different DAA-control modes will be dealt with in case of PAN membrane with adding DAA after feeding. One way of the DAA-control modes is to use DAA as the active agent to inhibit the biofilm formation for retarding membrane biofouling; the other is to serve DAA as the cleaning agent to break down the formed biofilm. As shown in Figure 3e, it can be obviously seen that the slope of the TMP curve was steeper when using DAA as a cleaning agent, which meant that a rapid membrane biofouling occurred on the MBR with DAA as a cleaning agent. However, it should be noted that the recovered initial TMP in the second and third cycle was lower than that of the MBR with DAA as an active agent, demonstrating that DAA played an important role in breaking down the formed biofilm on the membrane surface. Nevertheless, when DAA was served as the active agent, it preferred to control membrane biofouling and reduced the membrane biofouling rate by 6.96%, compared with the DAA control mode of using as the cleaning agent, thus effectively retarding the membrane biofouling, which was also supported by the total EPS amount represented in Figure 3f, with 25.90% lower EPS amount for the MBR with DAA as an active agent than that for the MBR with DAA as a cleaning agent. In addition, there were several advantages, such as simple operation, lower mechanical damage, etc., when DAA was employed as an active agent. Therefore, the DAA control mode of using as an active agent was expected to bring the MBR closer to possible beneficial practical application with reduced operational costs and efficient membrane biofouling control.

Overall, the aforementioned DAA control ways, such as membrane property, DAA-adding timing, and DAA-control mode, have significant influences on anti-biofouling performances of MBR. As we can see, the optimal DAA control ways were obtained by experiment, involving the hydrophilic membrane and DAA used as an active agent, as well as the DAA-adding timing of 4 h cultured before running. However, it should be noted that the time to reach the limit TMP (30 kPa) is shorter as the order progresses, which might be due to the complicated feeding composition with organic and inorganic components, resulting in combined fouling occurred in membranes. Nevertheless, DAA presented efficient anti-biofouling performances as a kind of biofilm inhibition agent. Simultaneously, the extent of their influence and interactive effects on membrane biofouling control needs to be further precisely and quantitatively optimized in order to more efficiently mediate membrane biofouling.

### 3.3. Optimization of DAA Control Ways for Efficient Alleviation of MBR Membrane Biofouling

Mathematic relationships between anti-biofouling performance and DAA control ways can be built, for precisely and quantitatively analyzing the weight of DAA control ways and ascertaining their interactive effects, and further optimizing and predicting DAA control way combinations to efficiently alleviate MBR membrane biofouling.

#### 3.3.1. Clarify the Weight of Different DAA Control Ways by MLR Model

As mentioned above, there were a couple of factors affecting the anti-biofouling performance, such as membrane property, DAA-adding timing, and DAA-control mode. Hence, it is appropriate to select the MLR model for representing the relationship between anti-biofouling performances and DAA control ways. First, through statistical analyses, key DAA control ways for mitigating membrane biofouling were identified. Forward regression was then used to obtain the mathematical representation of the membrane biofouling rate $r_{p}$ and different DAA control ways. Afterwards, according to the independent variable regression coefficient table (Appendix A), the MLR model was built to express the relationship between $r_{p}\;$and the significant correlative DAA control ways as follows:(3)$$r_{p}=1.069+0.089\;x_{1}\;-0.075\;x_{3}+0.088\;x_{1}x_{2}-0.126\;x_{1}x_{3}+0.049\;x_{2}x_{3},$$
where $r_{p}$ is membrane biofouling rate (kPa·h^−1^); $x_{1}$, $x_{2},\;$and $x_{3}\;$refer to the membrane property, DAA-adding timing, and DAA-control mode, respectively; the data chosen to derive the functions were the representative data measured during the three operation cycles of MBR, as shown in Appendix A, which covers the complete situation.

According to the model built above, the coefficient of determination R^2^ was 1, indicating that 100% of the variations occur in membrane biofouling rate ($r_{p}$), which can be elucidated by the independent factors and their interactions (Appendix A). Meanwhile, as the analysis of variance (ANOVA) represented in Appendix A, the significance test (significance level α = 0.05) of all coefficients were less than 0.05, revealing that each DAA control way has a significant impact on the membrane biofouling rate. Besides, it should be noted that the last two columns in Appendix A are collinearity statistics, in which the tolerance and variance inflation factor (VIF) are reciprocal. The greater the tolerance, the greater the possibility of collinearity. Hence, it can be seen that this type of best-approximated function may be developed for MBR to express the relationship between DAA control ways and anti-biofouling performances.

Furthermore, Appendix A depicts the Pearson correlation coefficients of the major DAA control ways and their interaction on the membrane biofouling rate of MBR. As we can see, the control ways (membrane property, DAA-adding timing, and DAA-control mode) all exhibited very strong impacts on membrane biofouling rate ($r_{p}$), and the contribution rate of the different DAA control ways on controlling membrane biofouling followed, as the order: DAA-control mode ($x_{3}$) > DAA-adding timing ($x_{2}$) > Membrane property ($x_{1}$). Among them, the absolute value of Pearson correlation coefficient of DAA-control mode is 0.721 and higher than other factors, thus being considered as the major factor for mediating MBR membrane biofouling. In the meantime,$\;x_{2}$ also played an important role, although it is heavily dependent on the variables of membrane property ($x_{1}$) and DAA-adding mode ($x_{3}$), representing in the form of $x_{1}x_{2}$ and $x_{2}x_{3}$ in Equation (3). Besides, the interaction between membrane property and DAA-adding timing ($x_{1}x_{2}$) was thought to be a particularly more important factor, resulting from which the Pearson correlation coefficient of $x_{1}x_{2}$ was as high as 0.730, as shown in Appendix A. Therefore, the effects of DAA control ways and their interaction on membrane biofouling control obtain precise and quantitative description through developing MLR model, laying an important foundation for further optimization of DAA control ways combination.

#### 3.3.2. Clarify the Optimization of DAA Control Ways Combination by RSM Model

It is beneficial to optimize DAA control ways combination for more efficiently mediating MBR membrane biofouling. Delightfully, RSM can yield useful optimization of interactive effects with considerably fewer experimental runs [26]. Herein, the RSM was utilized to investigate the dynamic and continuous membrane biofouling process under the different DAA control ways and their optimal combination to mitigate membrane biofouling. Box-Behnken design (BBD)-based RSM model, built in the Design-Expert 8.0 software, is considered as a reliable method to conduct the analysis of diagnostic plots, including the normal probability plot of residuals and predicted versus actual values, to validate the adequacy of the model [31]. As shown in Appendix A, the normal probability plot of the studentized residuals is a good graphical representation for the diagnosis of data normality, and the data are well fitted with the line, indicating that the data were normally distributed in the model responses [32,33].

Based on the output of the Box-Behnken design (BBD)-based RSM model presented in Appendix A, the optimum combination of DAA control ways for alleviating the membrane biofouling was obtained, i.e., DAA used as an active agent, and the DAA-adding timing of 4 h cultured before running, as well as both hydrophilic and hydrophobic membrane. Delightfully, the optimum DAA control ways by RSM model were consistent with that obtained in Section 3.2, suggesting that the RSM model is reliable to optimize DAA control ways and their combination for controlling MBR membrane biofouling. Further, the RSM model can be used to predict the desired DAA control ways and their combination for realizing more efficient membrane anti-biofouling performances.

#### 3.3.3. Model Prediction for Comprehensive Evaluation of MBR Membrane Anti-Biofouling Performances

Based on the MLR and RSM models established above, MBR membrane biofouling rates $r_{p}$ under different DAA control ways were predicted, as demonstrated in Appendix A.

Further, the relationship between the simulated and the actual value is depicted in Appendix A. It can be seen that the data were roughly distributed on a straight line, indicating the accurate prediction and simulation ability of the established model. In addition, the cube of predicted values of $r_{p}$ is shown in Figure 4a, representing all predicted $r_{p}$ under different combination situations of DAA control ways. Some detailed simulation values under certain combined control ways are also given in Appendix A. Noticeably, the optimized combined control ways involved DAA used as an active agent, and the DAA-adding timing of 4 h cultured before running, as well as both hydrophilic and hydrophobic membrane, as shown in Figure 4b, agreeing with the experimental results obtained in Section 3.2, and resulting in an approximately 40.24% decrease in the membrane biofouling rate in comparison with the conventional MBR.

In brief, the employment of MLR and RSM can accurately predict the degree of membrane biofouling during the whole operation process in MBR, thus promoting the efficient anti-biofouling operation for the practical MBR process with DAA control ways.

## 4. Conclusions

Different membrane properties, DAA-adding timings and DAA-control modes were investigated to evaluate the membrane anti-biofouling performances of DAA by experiments. Subsequently, the multiple linear regression model was built to further analyze the weight of different DAA control ways, exhibiting the following impacts on membrane biofouling rate as the order: DAA-control mode > DAA-adding timing > membrane property. Then, the response surface methodology was employed to optimize the DAA control ways, obtaining the optimum combination of DAA used as an active agent and DAA-adding timing of before running and both hydrophilic and hydrophobic membrane, yielding an about 40.24% reduction in the membrane biofouling rate in comparison with the conventional MBR. DAA is an effective means to combat MBR membrane biofouling under optimized control ways combination and a promising avenue to efficiently alleviate membrane biofouling in practical membrane systems.

## 5. Patents

Xiaoyan Guo, Zhan Gao, Shougang Fan. An intelligent optimization method for improving the efficiency of D-amino acid in reducing MBR pollution. 26 December 2020.

## Figures and Tables

**Figure 1 membranes-11-00612-f001:**
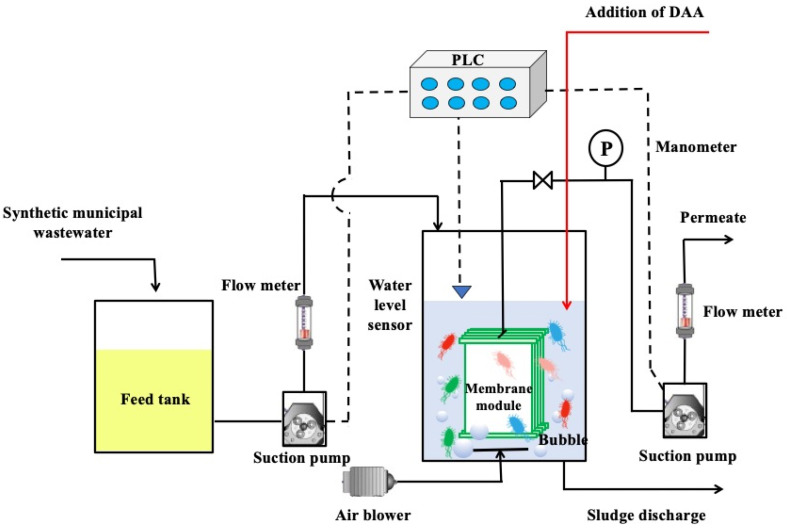
Schematic diagram of the MBR.

**Figure 2 membranes-11-00612-f002:**
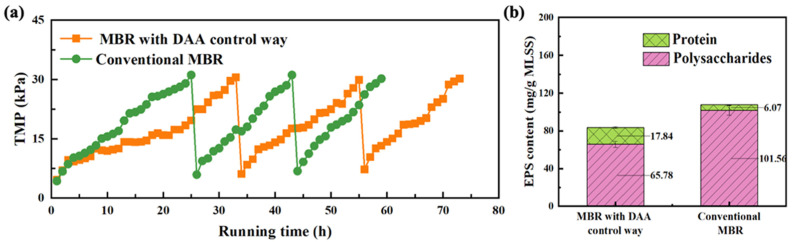
Evolvement profiles of membrane biofouling extent of MBR with DAA control way and conventional MBR. (**a**) TMP variation with running time for MBR with DAA control way and conventional MBR for three cycles. (**b**) Comparison of EPS production content between MBR with DAA control way and conventional MBR.

**Figure 3 membranes-11-00612-f003:**
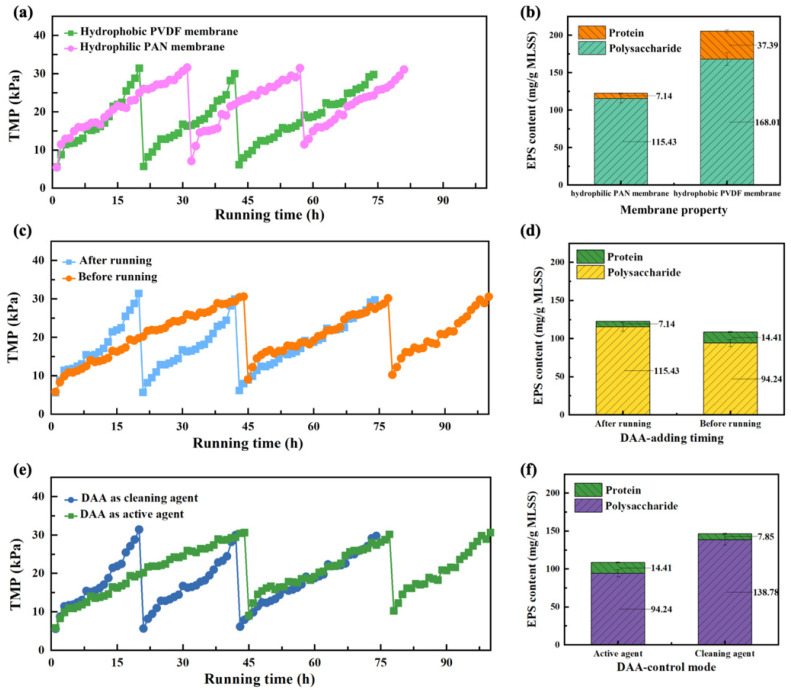
TMP variation with running time and EPS analysis for the MBR with different DAA control ways: membrane property (**a**,**b**); DAA-adding timing (**c**,**d**); DAA-control mode (**e**,**f**).

**Figure 4 membranes-11-00612-f004:**
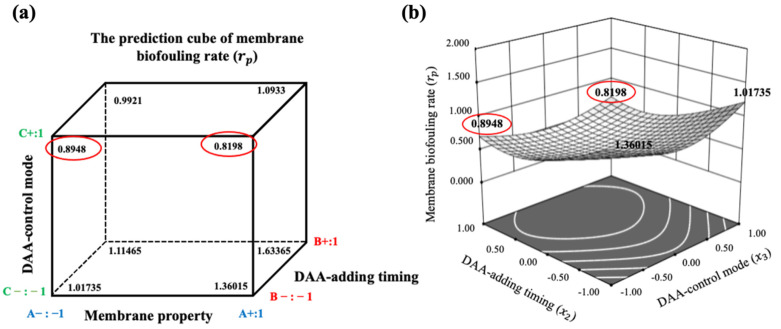
(**a**) The predicted values of membrane biofouling rate $r_{p}$ varied with membrane property (A), DAA-adding timing (B) and DAA-control mode (C) in cube. In which, A−: −1 for hydrophilic membrane; A+: 1 for hydrophobic membrane; B−: −1 for adding DAA before running; B+: 1 for adding DAA after running; C−: −1 for adding DAA as cleaning agent; C+: 1 for adding DAA as active agent. (**b**) The response surface plots showing the effects of DAA-adding timing and DAA-control mode on membrane biofouling rate.

**Table 1 membranes-11-00612-t001:** Properties of membrane modules tested.

Membrane Properties	Membrane Material	Effective Length (cm)	Membrane Fiber Quantity	Inner Diameter (mm)	Outer Diameter (mm)	Pore Diameter (μm)	Membrane Area (m^2^)
Hydrophilicity	Polyacrylonitrile (PAN)	10	320	0.7	1.2	0.2	0.1130
Hydrophobicity	Polyvinylidene Fluoride (PVDF)						

**Table 2 membranes-11-00612-t002:** The control protocols of DAA in MBR.

Control Protocols	Specific Information	
Membrane property	Hydrophilic PAN membrane	Hydrophobic PVDF membrane
DAA-adding timing	Before running (the DAA and microorganisms were cultured for 4 h * and then inoculated into the MBR for operation)	After running (adding DAA when TMP reached 15 kPa during the operation)
DAA-control mode	Active agent (adding DAA into the MBR before or after running)	Cleaning agent (after one biofouling-physical cleaning cycle, the membrane module was soaked in DAA solution of 6 mg·L^−1^ for 4 h *, and then pure water was used for rinsing)

* The more suitable time for effectively inhibiting membrane biofouling [16].

**Table 3 membranes-11-00612-t003:** Factors and levels of operating parameters in BBD.

Factors	Symbols	Levels	
		Low (−1)	High (+1)
Membrane property	$x_{1}$	Hydrophilicity	Hydrophobicity
DAA-adding timing	$x_{2}$	Before running	After running
DAA-control mode	$x_{3}$	Cleaning agent	Active agent

## Data Availability

Not applicable.

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
