# Peer review of "Exploration on Optimized Control Way of D-Amino Acid for Efficiently Mitigating Membrane Biofouling of Membrane Bioreactor"

_membranes, 2021, doi:10.3390/membranes11080612_

Round 1

Reviewer 1 Report

This research evaluated the anti-fouling performance of D-Tyrosine as bio-fouling reducers for MBR operation. The results of the study provide quite interesting and meaningful data that other researchers can be refer to. However, it seems that detail explanations or partial corrections are required to have better understanding of experimental results. 

  1. In Fig 3. three cycle were applied, would you explain why time to reach the limit TMP (30kPa) is shorter as the order progresses (especially PAN, Before running, DAA as active agents)?  From this point of view, doesn't continuous operation gradually reduce the DAA effect?
  2. Does the absence of an independent variable(x2) of "DAA adding time" in the MLR model of Equation 3 mean that the contribution is relatively small? further explanation is recommended.
  3. What is the mean of "VIF" at line 284?
  4. Membrane permeate flux  of 1.2 L·m-2· h-1 seems quite low. Do you have special reason to keep this flux? 
  5. At line 335-339, it is hard to find out how the 40.24% fouling rate reduction was calculated. Further explanation is also recommended how it agreed with experimental results obtained in section 3.2. 

Reviewer 2 Report

This is an interesting work assessing the biofouling mitigations by D-amino acid. The manuscript is clear and, in my opinion, the topic is of undoubted interest. I think that a significant contribution in the field of study. 

However, this manuscript is a preliminary study about the inhibition biofilm formation and disintegrating formed biofilm. In my opinion, a study about this fouling needs a long-term test, the authors using  short-term  test  (2-4 days). I suggest additional test to the conclusions are clearly supported by the results 

Round 2

Reviewer 2 Report

I thinks that long-term experiment is better that a prediction. However the conclusions are supported by the results.